# Mechanical Properties of Raw Filaments and Printed Specimens: Effects of Fiber Reinforcements and Process Parameters

**DOI:** 10.3390/polym16111576

**Published:** 2024-06-02

**Authors:** Daniel Vieweger, Sergej Diel, Hans-Georg Schweiger, Ulrich Tetzlaff

**Affiliations:** 1CARISSMA Institute of Electric, Connected, and Secure Mobility (C-ECOS), Technische Hochschule Ingolstadt, 85049 Ingolstadt, Germany; daniel.vieweger@tum.de (D.V.); hans-georg.schweiger@thi.de (H.-G.S.); 2Faculty of Mechanical Engineering, Technische Hochschule Ingolstadt, 85049 Ingolstadt, Germany; ulrich.tetzlaff@thi.de

**Keywords:** additive manufacturing, 3d printing, Fused Deposition Modeling, fiber reinforcements, process parameters, mechanical properties

## Abstract

Fused Deposition Modeling (FDM) is a well-established manufacturing method for producing both prototype and functional components. This study investigates the mechanical properties of FDM components by material and process-related influencing variables. Tensile tests were conducted on seven different materials in their raw filament form, two of which were fiber-reinforced, to analyze their material-related influence. To cover a wide range from standard to advanced materials relevant for load-carrying components as well as their respective variations, polylactic acid (PLA), 30% wood-fiber-reinforced PLA, acrylonitrile butadiene styrene (ABS), polycarbonate (PC), a blend of ABS and PC, Nylon, and 30% glass-fiber-reinforced Nylon were selected. The process-related influencing variables were studied using the following process parameters: layer thickness, nozzle diameter, build orientation, nozzle temperature, infill density and pattern, and raster angle. The first test series revealed that the addition of wood fibers significantly worsened the mechanical behavior of PLA due to the lack of fiber bonding to the matrix and significant pore formation. The polymer blend of ABS and PC only showed improvements in stiffness. Significant strength and stiffness improvements were found by embedding glass fibers in Nylon, despite partially poor fiber–matrix bonding. The materials with the best properties were selected for the process parameter analysis. When examining the impact of layer thickness on part strength, a clear correlation was evident. Smaller layer thicknesses resulted in higher strength, while stiffness did not appear to be affected. Conversely, larger nozzle diameters and lower nozzle temperatures only positively impacted stiffness, with little effect on strength. The part orientation did alter the fracture behavior of the test specimens. Although an on-edge orientation resulted in higher stiffness, it failed at lower stresses. Higher infill densities and infill patterns aligned with the load direction led to the best mechanical results. The raster angle had a significant impact on the behavior of the printed bodies. An alternating raster angle resulted in lower strengths and stiffness compared to a unidirectional raster angle. However, it also caused significant stretching due to the rotation of the beads.

## 1. Introduction

Additive manufacturing (AM) offers several technical, economic and environmental benefits due to its process-based principle of a tool-free, layer-by-layer component generation process. This is why the technology has gained popularity in various industries such as automotive, aerospace, and medical. The Fused Deposition Modeling (FDM) is one of the most well-known AM processes. FDM is the process of depositing molten thermoplastic material layer by layer through a movable extrusion nozzle onto a build plate [1]. FDM printers can process a wide range of materials, from beginner-friendly thermoplastics like PLA and ABS to more technical materials like PC and polyamide (PA), as well as high-performance plastics like polyetherimide (PEI) and polyetheretherketone (PEEK) [2,3]. Composites are often used to enhance and modify the material properties of polymers [4].

The FDM process involves several parameters that significantly affect manufacturing time, quality, weight, and mechanical characteristics [5]. Previous studies have extensively examined the impact of these parameters on the mechanical behavior of parts. One frequently studied parameter is the build orientation, which refers to the part’s orientation relative to the machine’s coordinate system. The orientation of the build is crucial as it heavily impacts the direction of the anisotropy [6]. Additionally, it determines the need and amount of support structures required, which in turn affects the post-processing and material consumption. Moreover, the orientation also affects the degree of the ‘staircase effect’, influencing both the dimensional accuracy and surface quality of the part [7]. Studies like Chacón et al. [8] and Kloke [9] suggest that selecting an orientation where the direction of deposition is parallel to the tensile load results in the most stable part. Conversely, the weakest strength is observed when the force acts parallel to the buildup direction.

The nozzle diameter is a rather rarely considered process parameter and describes the inner diameter of the nozzle. Typically ranging from 0.15 to 1 mm, larger diameters allow for more material to be extruded, resulting in time savings during the building process compared to smaller diameters. The nozzle diameter also impacts surface quality and mechanical properties [1]. Yang et al. [10] and Kiński [11] conclude that larger nozzle diameters result in an increase in tensile strength, a decrease in manufacturing time, and a decrease in surface quality. The increase in strength is said to be due to the smaller number of layers required for the same part height. Fewer layers reduce the number of necessary cooling and heating cycles, which can cause warpage, delamination, and manufacturing defects. However, Yang et al. [10] also achieved a 19% increase in tensile strength and a reduction in manufacturing time from 3.81 h to 1.30 h even with the same layer thickness by changing from a 0.2 mm nozzle to a 0.6 mm nozzle. This change resulted in a significant decrease in surface quality.

The layer thickness refers to the height of the deposited layer in the build-up direction, typically ranging from 0.05 to 0.4 mm. It directly affects production time and surface quality and reducing it can minimize the staircase effect [1]. However, studies on the influence of layer thickness on mechanical properties have yielded somewhat contradictory results. For instance, Huang et al. [12] found that the tensile and flexural strength decreases as the layer thickness increases. This is due to the increase in gaps between the layers, which leads to more defects in the component and ultimately results in premature failure. In contrast, Alafaghani et al. [13] and Yang et al. [10] did not observe this trend. Ladder showed that an improvement in mechanical properties is obtained with increasing layer thickness. According to this study, this is due to the reduced number of layers required, which leads to fewer deformations and cracks, thereby increasing the strength. However, Sood et al. [14] conclude that the tensile strength first decreases and then increases as the layer thickness increases. The study indicates that increasing the number of layers results in a higher temperature gradient towards the bottom of the component, which enhances the bonding of the adjacent raster. However, similar to the findings of Yang et al. [10], layer thicknesses are associated with deformation, cracking, and manufacturing defects. In addition, Chacón et al.’s [8] study also supports these observations. According to their results, increasing the layer thickness has a positive effect on the mechanical behavior of test specimens oriented in the Z direction, but not for flat or on-edge-oriented specimens. In the case of flat or on-edge-oriented specimens, the layer thickness had no significant influence.

The deposited bead’s width is referred to as the raster width, which is determined by the nozzle diameter, flow rate, and printing speed. The angle of the deposition direction to the X-axis of the printer is known as the raster angle. According to Kloke’s [9] research, the mechanical properties of the test specimens are minimally affected by the width. However, slightly better results were obtained with wider raster strings. The fracture and mechanical behavior are significantly affected by the raster angle. Kiendl and Gao [15] conducted a series of tests that demonstrated a decrease in Young’s modulus and strength values with increasing unidirectional raster angle. In this context, 0° refers to the orientation of the raster in the direction of loading. When the tensile load is parallel to the direction of the deposited beads, the pure material strength is approximately achieved. However, when the raster angle increases (>0°), the strength results from the angle of the tensile loading and the strength of the bonds between adjacent beads [9]. If the raster angle is built up alternately, for example, ±45°, there is an increase in elongation at break with a reduction in strength. Kiendl and Gao [15] argue that the individual rasters rotate towards the loading direction before failing. Consequently, the mechanical behavior can be changed from brittle to more ductile based on the raster angle.

To save time and material, FDM parts are often produced with an internal structure rather than being solid. This inner structure is defined by the infill pattern and the infill density [1]. In tests on different infill densities, Abeykoon et al. [16] showed that the Young’s modulus of the tensile test specimen at an infill density of 100% was almost twice as high compared to the same specimen with a 25% infill density. Furthermore, the study demonstrated that the linear infill pattern had the highest Young’s modulus at a 50% infill density, while the hexagonal pattern had the lowest. Algarni and Ghazali’s research [3] also indicates that the best mechanical properties are achieved at a 100% infill density when considering different materials. Harpool [17] and Moradi et al. [18], on the other hand, state that the hexagonal structure performs best and by using an optimum combination of infill density and pattern, better values than with a solid specimen can be obtained. It is important to note that these are isolated study results. The general consensus in the literature is that the tensile strength increases with increasing infill densities [1]. In addition to mechanical properties, factors such as cost, time, and weight must also be considered in the selection process. For instance, Abeykoon et al. [16] found that the mass of a tensile test specimen increased by 68% when the infill density increased from 25% to 100%. Therefore, it is recommended in various studies and the literature that the mechanical properties of the component should be determined first. Then, the appropriate infill density and pattern can be selected to save weight, cost, and time [16,19,20].

The temperature at the nozzle exit, known as the nozzle temperature, affects the mechanical properties of the extruded material by altering its viscosity, velocity, and flowability [1,20]. According to Akhoundi et al. [21], increasing the nozzle temperature improves bonding between the rasters and layers, resulting in increased tensile strength and Young’s modulus. However, it is important to note that if the temperature is too high, the bond between the two layers may deteriorate if the solidification of the deposited material has not progressed far enough while a new layer is already being applied [16]. Therefore, it is crucial to match the printing speed precisely with the nozzle temperature. In most cases, the recommended temperature range is specified by the material manufacturers. This parameter has not yet been extensively investigated. 

The relationships between individual process parameters are complex and non-linear and are not yet fully understood. Therefore, it is not surprising that different studies and literature present varying results and conclusions. Standardizing the results of different research studies is challenging due to the non-standardized processes, materials, and test procedures used in the FDM process. All studies emphasize the significant impact of process parameters on mechanical properties, surface quality, and manufacturing time. While the effects on surface quality, mass, and manufacturing time are evident and can be explained simply, more complex relationships arise for mechanical behavior. Accordingly, this work is intended to complement and extend previous research in order to contribute a part to a broader and deeper understanding of AM. 

The investigations are based on tensile tests to determine the basic material parameters strength, stiffness, and elongation at break, which are particularly important for component design in the early stages of development. Unlike previous research, the tests are carried out under the same conditions to ensure comparability of results. The tests are split into two series. In addition to previous investigations, the first series examines the material properties of seven raw filaments, two of which are reinforced with different fiber materials, to investigate the transferability of the raw filament properties. The choice of materials has been made to cover a wide range from standard, beginner-friendly polymers to more advanced engineering materials, as well as their respective developments and variations to improve their properties. This is why a composite with recycled wood fibers was included, as the use of recycled materials in filaments has recently been a significant research focus to enhance sustainability in AM [22,23]. The first test series allows good comparability of each material and serves as a pre-selection of materials with the best mechanical properties for the investigation of process parameters. The selected parameters, which include layer thickness, nozzle diameter, build orientation, nozzle temperature, infill density, infill pattern, and raster angle, represent a comprehensive range of variables central to the FDM process. For both test series, SEM images of the fracture surface from raw filaments and printed specimens are taken to gain a better understanding of the microstructure and its relationship to the mechanical properties. In addition, the weight-related properties are analyzed in order to better exploit the lightweight construction potential of structural components. Based on the information about the mechanical properties of the materials and the influence of the process parameters, real components can then be designed and optimized.

## 2. Materials and Methods

### 2.1. Materials and Experimental Setup

This study compares seven materials, including two fiber-reinforced variants, in their original filament form. The materials selected for the first group include pure PLA and PLA with recycled wood fibers. PLA is widely used in FDM due to its simple printability and relatively good mechanical properties. Both filaments are in the rather lower price segment and belong to the group of bioplastics. Furthermore, the materials ABS, PC and a polymer blend of ABS and PC are considered. These filaments belong to the group of engineering plastics due to their good mechanical and thermal properties. The blend, in particular, benefits from the synergistic effects of the individual components. The last two materials are PA6 and glass-fiber-reinforced PA6. The more common name for PA is Nylon, which will also be used in the following. Accordingly, the glass-fiber-reinforced PA6 is referred to as Nylon + GF. For more detailed information on these materials, please refer to Table 1. The information presented in the table is taken directly from the technical data sheets provided by the manufacturer.

The materials are processed using two FDM 3D printers: the Prusa i3 MK3S from Prusa Research and the Ultimaker2+ from Ultimaker. The Prusa Research printer requires filaments with a diameter of 1.75 mm and does not have an enclosed build space. The slicing software developed by Prusa Research is called PrusaSlicer. The Ultimaker2+ works with 2.85 mm thick filaments and allows printing in a closed environment due to additional housing. The software used is Ultimaker Cura.

The mechanical properties investigation comprises two series of tests. The first series aims to measure the material properties of the raw filaments listed in Table 1. This resulted in a total of 21 tests for the first series, using three specimens per material. The filament’s tensile tests were performed using the Instron Electropuls E3000 tensile testing machine with a 3 kN load cell. The strain was calculated from the ratio of the change in length to the travel distance of the machine, as this machine does not have a strain gauge. This type of strain measurement works well in this case due to the relatively compliant filaments, and the machine’s own deformation can be neglected. The cross-sections of the specimens were determined based on the respective filament diameters. The filament pieces were 170 mm in length, and the test speed was set to 2 mm/min. With 35 mm long clamps, this results in a free clamping length of 100 mm. The filaments maintain a constant thickness, which means that failure at or within the clamps is expected for all materials due to the additional clamp stress. Therefore, only the yield strength and the tensile modulus can be evaluated. Other mechanical properties, such as tensile strength or elongation at fracture, would be meaningless and are therefore not assessed. Since no appropriate test standard has been found for the tests on filaments, the tests were based on the standard DIN EN ISO 527-3 [31].

The second series analyzes the influence of various parameters using printed specimens. A total of 51 tests were conducted using 17 different configurations, with three samples tested per configuration. The DIN EN ISO 527-2 standard [32] is used as a reference for the testing of the printed specimens. The tests were conducted using the Instron 5966 universal testing machine with a 10 kN load cell. The optical gauge records the strain based on two markings on the specimen. All experiments are conducted under standard room conditions.

There is currently no standardized tensile test specimen for AM components. According to Kloke [9] and Knoop [7], the test specimen specified in the DIN EN ISO 527 standard has an unsuitable geometry for production using the FDM process. This is because it has various weak points depending on the building’s orientation. Flat-oriented specimens show enlarged gaps in the radius region depending on the raster angle due to the bead deposition process, which leads to an increased notch effect. For specimens oriented on-edge, a support structure must be used, which should be removed mechanically afterward. However, this process can result in surface defects that may lead to crack initiation. Additionally, weak points may exist in the radius area due to the staircase effect, which could potentially lead to inaccurate test results. Figure 1 illustrates the two build orientations and the definition of the raster angle. Tests were conducted using various sample geometries to discover a more suitable geometry for reducing the notch effect. All geometries were designed in the CAD program CATIA V5 and printed using the Prusa i3 MK3S printer with PLA material. In addition to waisted specimens, various versions of rectangular specimens were tested according to DIN EN ISO 527-4 [33], as they do not have enlarged gaps. However, these specimens failed mostly in or at the clamps due to the added clamping force. Figure 2 illustrates the geometry which is most suitable based on preliminary tests. This specimen has a larger radius compared to the standardized specimen, which reduces the notch effect and shifts the process-related weak point due to gaps towards the center. However, due to the round nozzle geometry, it is not possible to completely avoid gaps. To reduce the weak points, the overlap of the infill geometry with the walls was increased.

Table 2 presents the experimental plan, and Table 3 displays the parameters to be investigated. The study aims to guide the manufacture and design of real parts. Therefore, only materials with the best mechanical properties from the first test series and with the best printability, namely ABS + PC, Nylon + GF and PLA, will be investigated further. The choice of material for each test was adapted to the conditions of the material and the 3D printers. For example, PLA was used to investigate the different nozzle temperatures, as it was only possible to exceed the recommended temperature with PLA due to the maximum nozzle temperature of the 3D printers. As another example, Nylon + GF was used to study the influence of different raster angles since the effects of shear forces on the fiber–matrix bonds are of high interest for composites. With these conditions considered, the test plan was designed to investigate all parameters with the minimum number of samples. The test speed is set to 2 mm/min, based on similar studies (1–5 mm/min) [8,13,16,17]. Three specimens are produced and tested for each design variant (ID). The test specimens with different infill densities were previously weighed using a precision balance. Later, the masses will be related to the measured mechanical values. The test plan was adjusted multiple times due to complications in the manufacturing process. Some specimens could not be manufactured, resulting in their removal from the test plan. As the IDs were assigned before manufacturing, some letters are missing in the following table (Table 2).

### 2.2. Preparation and Execution

All materials must be subjected to the same environmental conditions before and during testing. For materials that absorb moisture quickly, it is recommended or prescribed to store them in a dry environment. Therefore, all materials were stored in a dry box from the beginning. The filaments were only removed for the processing procedure. The dry box is a standard plastic box with a volume of 44 L, which was sealed and filled with approximately 2 kg of desiccant (silica gel). For the first series of tests, the filaments were then cut to the specified length and tested using the tensile testing machine (Figure 3a). Prior to producing the test specimens, it is necessary to dry the filaments with a high hygroscopy in accordance with the manufacturer’s instructions. As such, these filaments were dried at approximately 70–80 °C for 2 h in an oven. Following the material specifications outlined in Table 1, the 3D printers were then prepared, and the manufacturing process was initiated using the data generated in the respective slicing software. The specimens were then post-processed according to their build orientation. Reference points with a distance of 50 mm were centrally painted on the bodies for the optical strain gauge of the tensile testing machine (Figure 3b).

## 3. Results and Discussion

### 3.1. Tests on Filaments

Table 4 shows the average values and corresponding standard deviations of the measured mechanical properties for all seven materials. As anticipated, stress concentrations caused all materials to fail in or near the clamping area. Therefore, only the yield strength σ_y_, which is the stress at 0.1% plastic strain, and Young’s modulus E will be evaluated. Comparing the measured Young’s modulus values with the values given in Table 1, it is obvious that there are some large differences. Since important information about the test methods is missing from the data sheets and a different test method was used, any explanation of these differences would be speculative. Therefore, this discrepancy is noted but cannot be further explained.

During the tests and upon examining the damaged specimens, it was observed that PLA, ABS, PC, ABS + PC, and Nylon exhibited clear necking and ductile behavior. In contrast, the fiber-reinforced materials all failed due to brittle fracture (Figure 4).

Figure 5 summarizes the yield strength and Young’s modulus values. PLA and PC exhibited almost identical strength values, although the PC tests showed comparatively high variations. The polymer blend had slightly lower strength, with a reduction of approximately 5 MPa. The next lowest value was achieved by the unreinforced Nylon. The filaments with the lowest yield strengths are PLA + Wood and ABS, both at about 20 MPa. Glass-fiber-reinforced Nylon achieved the highest yield strength. A similar trend can be observed for stiffness evaluation, as shown in Figure 5b. Nylon + GF has the highest Young’s modulus at about 4.700 MPa, while PLA has a stiffness of about 2.000 MPa less. ABS + PC has the next lowest value, followed by PLA + Wood, Nylon, and PC. ABS has the lowest stiffness.

When comparing the values of PLA with the PLA + Wood, it is evident that the recycled wood fibers function more as a sustainable filler than a reinforcing component. Both strength and stiffness significantly deteriorated due to the use of wood fibers. The yield strength loss was 50%, and the stiffness loss was 36%.

Figure 6a presents the stress–strain curves of PLA and PLA + Wood. PLA exhibits visible necking with a stress drop after reaching the tensile strength, while the wood-fiber-reinforced PLA fails due to its brittleness shortly after reaching the maximum stress.

Figure 6b illustrates the presence of large pores in the PLA + Wood material, distributed throughout the fractured surface, with a diameter of up to 50 µm. The origin of these pores remains unclear. The cellular structure of the material and the resulting stress peaks during loading lead to a significant reduction in stiffness and strength. The gaps between the wood fibers and the surrounding PLA suggest a lack of connection between the constituents. Due to the less-than-optimal adhesion of the wood fiber, it can be assumed that they do not improve the mechanical properties to any great extent, but rather serve as a placeholder. Kuciel et al. [34] have shown that the addition of wood fibers results in a significant reduction in tensile strength and elongation at break. However, in contrast to the present study, they found that stiffness increases with increasing wood content. The difference in behavior observed in the present study may be attributed to the cellular structure of the material, as shown in Figure 6b.

The strength values of ABS, PC and ABS + PC indicate a slightly negative influence of the rather weak ABS on the PC, as the yield strength of the pure PC is higher than that of the blend. However, the difference in values is minor. On the other hand, the stiffness analysis shows a synergistic effect between the individual components, resulting in a 19% increase over the PC. The stiffness increase compared to the ABS is 43%. Based on the given values, an increased PC content is assumed. Figure 7 displays exemplary stress–strain curves of the three materials.

Compared to wood fibers, the use of glass fibers significantly improves the mechanical properties of Nylon. The glass fibers have significantly higher mechanical properties than the polymer matrix, resulting in a 70% increase in yield strength and a 160% increase in stiffness for the composite. Figure 8a displays the recorded stress–strain curves. Nylon exhibits a ductile behavior without a pronounced stress drop, while the fiber-reinforced Nylon breaks immediately after reaching the tensile strength. The preferred orientation of the fibers along the applied force can be observed in Figure 8b, which also contributes to the strengthening effect. The glass fibers have a diameter of approximately 15 µm. Despite the mechanical values, SEM images of the fractured surface show partially poor bonds between the fibers and the matrix (Figure 8b). This may be due to glass fibers being pulled out during the tensile tests. No SEM investigations have been conducted on untested filaments, and therefore no statement can be made as to whether the fiber–matrix bonding is optimal or not.

Three materials with the highest strength and stiffness, namely PLA, ABS + PC, and Nylon + GF, were selected for further investigation on printed specimens based on the results of filament testing. Table 2 and Table 3 summarize the materials and process parameters investigated.

### 3.2. Tests on Printed Specimens

Table 5 presents the average values and standard deviations of all specimens’ measured values. As previously stated, comparing the measured values of this work with the values in Table 1 is unreasonable due to the differing test procedures and setups, such as different test speeds, the use of injection-molded rather than printed specimens, and different process parameters.

The test results will be compared and evaluated in the following sections according to the experimental plan. The aim is to demonstrate the influence of different process parameters.

#### 3.2.1. Effect of Layer Thickness (A, B, C)

Figure 9 presents the results of test specimens A, B and C, indicating that lower layer thicknesses can achieve higher strengths. For specimen C, only one test was conducted until failure, hence the elongation at break scatter could not be calculated for Table 5. Necking mostly occurred in the transition areas between the taper and the rectangular section. Due to the lateral build orientation, this corresponded to the end or initial regions of the staircase effect (Figure 10).

Figure 9a shows a clear correlation between yield strength and layer thickness. The specimen with a layer thickness of 0.1 mm achieved the highest yield strength of 57 MPa, while the specimen with a layer thickness of 0.3 mm had the lowest strength of 45 MPa. This confirms the findings of Huang et al. [12] that an increase in layer thickness results in a higher degree of defects and lower strengths. Increasing the thickness of each layer leads to a larger bead diameter because the extruded material is placed on the previous layer with less pressure. This results in increased gaps between layers and beads. In addition, the process-related staircase effect and roughness increase. The oblique necking of the bodies and the values indicate that the strength of the on-edge built-up bodies depends heavily on the strength of the layer bonds. Furthermore, it is clear that the filament has a lower yield strength than the printed specimens.

The stiffness is presented in Figure 9b. Unlike the strength, there is no clear dependency on the layer thickness. The Young’s modulus of the filament is within the same range as that of the printed specimens.

#### 3.2.2. Effects of Nozzle Diameter (C, D)

Changing the nozzle diameter from 0.4 mm to 0.6 mm did not significantly affect the strength of the specimens, but it did affect their stiffness. Specimen D behaved similarly to the other on-edge-oriented specimens. The tensile strength improved by 2 MPa when using a larger nozzle diameter, while the yield strength remained the same for both variations at 45 MPa. However, the stiffness increased by over 300 MPa to 2.340 MPa. The part exhibited a slightly more brittle behavior. The stress–strain diagrams in Figure 11a show the different behaviors.

The behavior may be attributed to the larger diameter, which reduces the heating and cooling cycles even with the same layer thickness. Yang et al. [10] found that more cycles lead to distortions and manufacturing defects and, therefore, to weaker mechanical properties. Additionally, the thicker beads next to and on top of each other result in larger contact areas. Due to the lower number of beads in specimen D, its mechanical behavior is less influenced by individual bead connections and more by the material properties (Figure 11b). As a result, the production time for specimen D was reduced from 52 min (as for specimen C) to 30 min, which represents a reduction of approximately 40% while simultaneously improving the mechanical properties.

#### 3.2.3. Effects of Orientation (E, G)

When comparing the recorded values of the two different orientations, it is important to take into account the significantly different behaviors observed during testing. Test specimen G failed uniformly and suddenly over the cross-section, while isolated layered composites of test specimen E cracked sequentially. Additionally, individual composites in the body failed well before reaching the tensile strength. The stress–strain diagram for test specimens E and G clearly shows a difference (Figure 12a).

Therefore, it is important to consider the strength values in relation to the stress–strain curves’ progression. Using the maximum stress values achieved as the reference point, the on-edge-oriented specimen reaches slightly higher values of 60 MPa. However, this is technically irrelevant due to the first signs of failure occurring well before the tensile strength. Comparing the yield strength, it becomes clear that the flat orientation is to be preferred. For specimen E, the highest values before the first stress drop were used as the yield strength. On average, this value is 19 MPa lower than the value for the flat orientation. For stiffness, the on-edge orientation achieved better results. The average value of specimen E at 4.339 MPa was almost 700 MPa higher than for specimen G. It is unclear whether and to what extent the different weak points of the different orientations due to the bead deposition procedure of FDM had an influence on the results.

Figure 12a shows that the first components of specimen E failed at approximately 33 MPa. This early failure, likely due to surface defects and inaccuracies resulting from the necessary support structure, must be considered in the design of real parts. The crack progression clearly indicates the failure of individual areas of the body, which tended to run between the layers, in contrast to the crack in the flat-oriented body. The strength of the composite layer depends on the thickness and area of each individual layer. Based on the test results with different layer thicknesses, the strength should increase by reducing the layer thickness and increasing the layer area. Specimen E exhibited a more unpredictable failure behavior compared to specimen G. In the case of on-edge built specimens, the crack likely initiated at the taper on the side where the support structure was attached and propagated mainly along the bead deposition direction towards the opposite side. Figure 12b illustrates the contrast in fracture behavior between the two orientations.

#### 3.2.4. Effects of Nozzle Temperature (I, J, L, M)

The results of test specimens I, J, L, and M, presented in Figure 13, indicate that the nozzle temperature had a minimal effect on the mechanical behavior of PLA. The strengths of the specimens printed at different nozzle temperatures were nearly identical. It is worth noting that the strength of the filament is significantly lower than that of the printed specimens (Figure 13a). Regarding stiffness, Figure 13b shows a clear correlation between nozzle temperature and stiffness.

As the nozzle temperature decreases, the stiffness of the component increases, but only slightly. The difference in values between a nozzle temperature of 230 °C and 190 °C is 259 MPa. However, these values contradict the results recorded by Akhoundi et al. [21]. Figure 14 displays the SEM images of the fractures of the specimens printed with nozzle temperatures of 190 °C and 230 °C. Both specimens exhibit a brittle fracture pattern. Additionally, the specimen with a nozzle temperature of 230 °C shows a partial interlaminar failure. In general, the damage pattern indicates that the interlaminar strength is sufficient, as no cracks are discernible between the individual strands and layers. The characteristic voids between the beads or layers are clearly visible. As noted by Akhoundi et al. [21], these voids are smaller at higher temperatures, resulting in better connections. The previous results indicate that stiffness may improve, but the effect is small for specimens loaded in the direction of bead deposition. The correlation observed in this study is likely due to differences in the degree of crystallization during the cooling of the beads. It appears that the degree of crystallization increases with stiffness at lower temperatures. It is possible that the first polymer chains decomposed due to the higher temperatures. Additionally, the bodies exhibit significantly different standard deviations (Table 5). Higher nozzle temperatures appear to result in larger stiffness variances. To assess the impact of bead connections, specimens with a raster angle α ≠ 0° must be selected and tested. Similarly, tests on specimens built up in the Z-direction can help evaluate the influence of layer connections.

#### 3.2.5. Effects of Infill Density (F, N, O, P)

The study confirmed the correlation between infill density and mechanical behavior. Lower infill densities resulted in reduced strengths and stiffness of the PLA bodies. Specifically, a reduction in infill density from 100% to 40% led to a 40% reduction in strength and a 35% reduction in stiffness. When comparing the filament to specimen F (100% infill density), it is evident that while the stiffness values are similar, there is a notable difference in the strength values (Figure 15). While the elongation at break is relatively constant for samples F, N and P and the samples show rather brittle behavior, sample O shows slightly more ductile behavior with slightly higher elongation at break, which is accompanied by greater scattering (Table 5). This behavior cannot be explained on the basis of the test program alone. Further investigations with the aid of SEM could possibly provide more insight.

When designing lightweight components using AM, it is important to consider the relationship between their values and respective densities to obtain specific mechanical values. To calculate densities, the previously recorded masses were divided by the volume of the specimen geometry, which was calculated in CAD (Table 6). The density of the filament used can be found in Table 1.

As previously stated, FDM’s process principle makes it impossible to produce completely solid bodies, resulting in an internal structure with voids, as shown in Figure 14. Therefore, a 100% infill density does not achieve the manufacturer’s specified material density of 1.24 g/cm^3^, but rather 1.23 g/cm^3^. Figure 16 illustrates the relationship between yield strength, Young’s modulus, and density.

Figure 16 shows that the strength and stiffness values remain nearly constant regardless of the infill density, indicating a linear decrease in mechanical properties with density. It is worth noting that this relationship is typically non-linear for cellular materials, making the linear relationship found here the upper limit for material properties. The stiffness and strength of cellular materials are significantly lower, as noted by Gibson and Ashby [35]. This characteristic allows for significant advantages in lightweight construction to be achieved in practice.

#### 3.2.6. Effects of Infill Pattern (N, Q, R)

The mechanical behavior of components was clearly affected using different infill patterns at a 40% infill density. Specimen N, which had a rectilinear infill pattern along the X-axis, fractured in the initial area of the taper due to gaps forming between the infill and the walls as a result of the process principle. This fracture behavior has already been explained earlier. In contrast, specimen Q tended to fail in the measuring range. The honeycomb pattern’s unique structure explains this phenomenon. Unlike the other two patterns, it consists of three repeating orientations, as shown in Figure 17.

This alternating structure ensures that the failure points in the edge area are reduced in size and covered. Consequently, the failure point tends to shift to the intended measurement area in the center. According to a comparison of the tested specimen R with the slicing program, the grid structure also has an unintentionally predetermined weak point outside the measuring range. This weak point is located where the grid structure is not connected. Additionally, the identified areas of weakness in the deposition alignment along the X-axis are situated at the initial and final layers, as shown in Figure 18.

The test results can be explained based on the respective structures. The linear infill pattern exhibited the highest strength with an average of 37 MPa, followed by the honeycomb structure with 31 MPa, and the grid structure with the lowest strength of 30 MPa (Figure 19a). It is worth noting that the fracture occurred at a significantly larger cross-section for the grid structure, which may indicate that the strength value needs to be adjusted downwards.

The stiffness results were consistent with the strength results, with values of 1.839 MPa for the linear pattern, 1.725 MPa for the honeycomb pattern, and 1.635 MPa for the grid pattern. The fundamentally better behavior of the linear pattern results from the complete orientation of the beads in the direction of loading. Thus, the values are mainly influenced by the material properties of the PLA. The grid structure also has this orientation of the beads, but they are significantly fewer in number. The 90° beads do not bear much load during tensile loading and are considered interference points. This results in poorer mechanical behavior. Based on the structure of the honeycomb, it can be seen that the mechanical behavior is mainly determined by the bead connections. The weaker connections between the beads result in a weaker component compared to the material properties. When subjected to loads transverse to the X-axis, the honeycomb and grid structure are expected to produce better values due to their significantly lower anisotropy.

#### 3.2.7. Effects of Raster Angle (G, S)

As with the specimens in the previous chapter, those with a raster angle of 0° broke at the expected area, while those with a raster angle of ±45° always failed within the measurement range. This is because the defects in the radius are much smaller and each layer covers the gaps on the opposite side (Figure 20).

It can be assumed that using an alternating raster angle instead of a unidirectional raster angle may result in a higher real infill density. The specimens with different raster angles behaved as described in the literature [1,15]. The strength and stiffness decrease while the ductility increases when the part is built with an alternating raster angle. Quantitatively, the yield strength decreased by approximately 20%, and the stiffness decreased by approximately 34%. The elongation increased by more than two times, resulting in a more ductile behavior. Figure 21 illustrates a stress–strain diagram showcasing the behavior dependent on the raster angle.

Based on the various gradients, it is evident that the beads deposited alternately must first align themselves in the direction of the load, resulting in significantly lower resistance to deformation. The mechanical behavior is less influenced by the material properties and more by the properties of the bead connections. However, the strength loss remains below expectations or even lower than reported in the literature. One possible explanation for this phenomenon is the reduction in the gaps between the beads caused by the insertion of fibers. This phenomenon was previously observed by Tekinalp et al. [4] in their studies on carbon-fiber-filled polymers. The good thermal conductivity of carbon is cited as the reason for this. The SEM images of the damaged specimen confirm this phenomenon for glass fibers despite their much lower conductivity compared to carbon (Figure 22). Even with SEM imaging, it is not possible to differentiate between individual beads. The complete fusion of the beads enhances the strength of the bead connection and, consequently, the strength of the entire component. The images also reveal the brittle, interlaminar fracture behavior of the bodies. The individual layers can be clearly distinguished from each other, particularly in the specimen with the alternating ±45° raster angle. Consequently, the fibers’ orientation according to the deposition direction is clearly visible. The fibers bond well to the Nylon matrix, which was not the case with the unprocessed filament (Figure 23). This is likely due to the additional pressure applied when extruding the melted filament through the small nozzle opening. The connection between the individual layers was not explicitly examined. However, the damage pattern indicates that there is sufficient bonding, as no major detachment was recognizable for tested specimens.

## 4. Conclusions

In this study, material-related and process-related influencing variables were investigated in two test series. The first test series was used to identify the most suitable materials the second test series. The materials tested in their unprocessed filament form include PLA, PLA + Wood, ABS, PC, ABS + PC, Nylon, and Nylon + GF. The results suggest that wood fibers should be used more as a filler than a reinforcing component. Compared to pure PLA, the mechanical properties of the material deteriorated significantly. This is due to poor or non-existent bonding of the wood fibers in the matrix and pore formation within the matrix, as evidenced by SEM images. The polymer blend of ABS and PC showed a slight increase in stiffness compared to the individual components, but the strength was slightly reduced due to the negative effect of the ABS properties on the material. The addition of a glass fiber insert resulted in significant increases in both the strength and stiffness of the Nylon material. Additionally, the fracture behavior changed from ductile to brittle due to the high fiber content of 30%. SEM images clearly show the orientation of the fibers. However, the weak bonding between the fibers and the matrix was unexpected.

The three materials PLA, ABS + PC, and Nylon + GF were selected for investigation of the parameters based on the results from the first series of tests. The second series of tests yielded the following findings:
The layer thickness exhibited a clear correlation with the strength of the components. Specifically, the smaller the layer thickness, the higher the strength. This is due to the fact that smaller layer thicknesses increase the pressure of the current layer on the previously laid layer, thereby reducing the gaps between the beads and the layers.The nozzle diameter had a minor effect on part behavior, but larger diameters increased stiffness due to larger contact areas and fewer heating cyclesThe fracture behavior of the flat and on-edge built specimens exhibited significant differences. The on-edge built part had a higher stiffness but showed early failure, making the flat orientation preferable.The different nozzle temperatures exhibited a correlation with the stiffness of the components. As the temperature increases, the stiffness of the specimens decreases due to varying degrees of crystallization. The SEM images indicate that higher temperatures result in smaller gaps, but this effect is less significant for loads along the bead deposition direction than the degree of crystallization.The infill density exhibited a clear correlation with the mechanical properties of the specimens. Specifically, the strength and stiffness decreased as the percent infill density decreased. Compared to the usual behavior of cellular materials, the mechanical properties only decrease linearly with density, which indicates a high potential for lightweight design.The evaluation of the three infill patterns revealed that the pattern-oriented completely in the loading direction performed the best. Strength and stiffness deteriorate as soon as the beads incline more than 0° to the loading direction.Both raster angles displayed the typical stress–strain curve. The 0° specimen had a steeper rise but broke at a lower strain compared to the ±45° specimen, which had a flatter curve but stretched further. SEM images demonstrate a significant improvement in the bond between fiber and matrix due to the processing procedure. The conductivity of the glass fibers results in a perfect fusion of the individual beads, making them indistinguishable from each other in the SEM images.

## Figures and Tables

**Figure 1 polymers-16-01576-f001:**
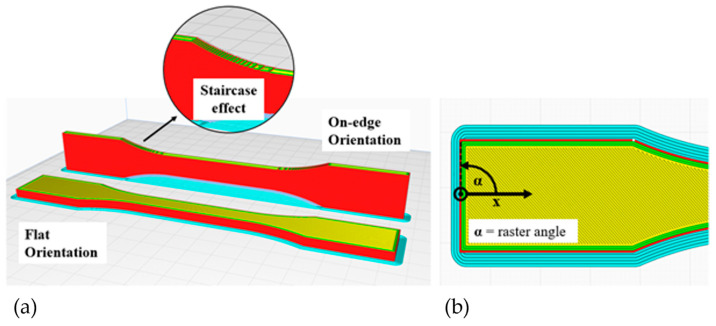
(**a**) Build orientations and (**b**) raster angle.

**Figure 2 polymers-16-01576-f002:**
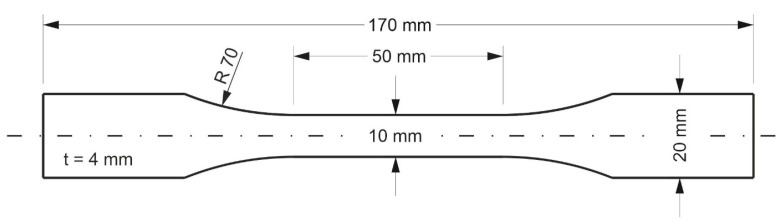
Specimen geometry for tests of printed specimens.

**Figure 3 polymers-16-01576-f003:**
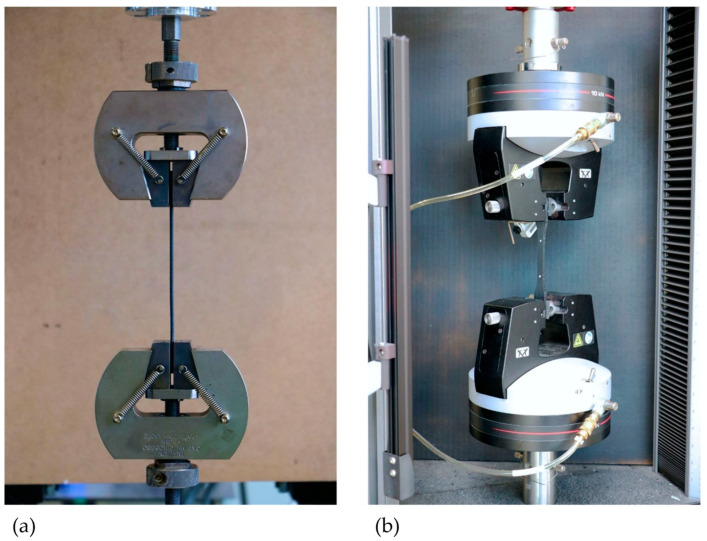
Experimental setup: (**a**) tests on filaments, (**b**) tests on printed specimens.

**Figure 4 polymers-16-01576-f004:**
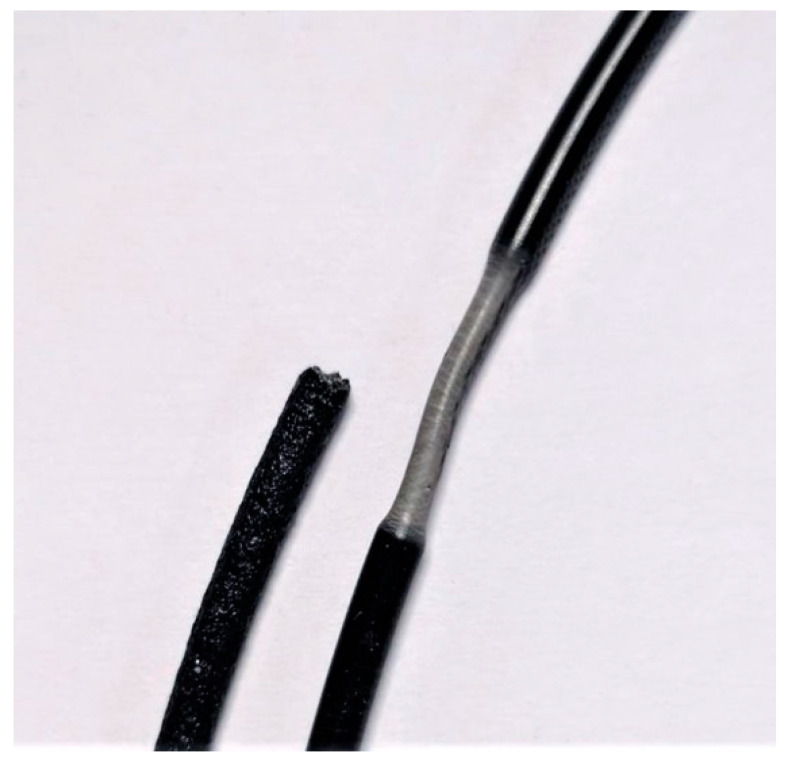
Comparison of a brittle (left, Nylon + GF) and a ductile (right, ABS) filament failure.

**Figure 5 polymers-16-01576-f005:**
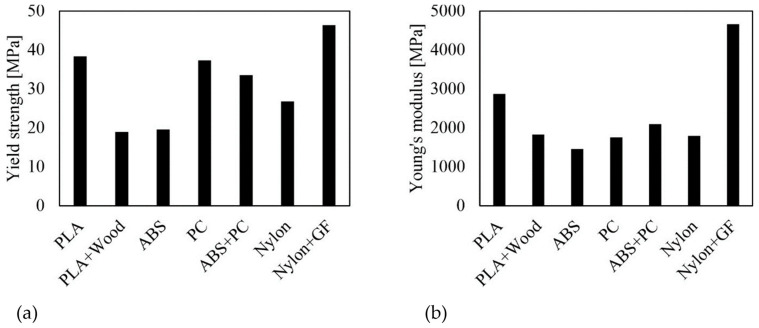
Mechanical properties of filaments (average values): (**a**) Yield strength and (**b**) Young’s modulus.

**Figure 6 polymers-16-01576-f006:**
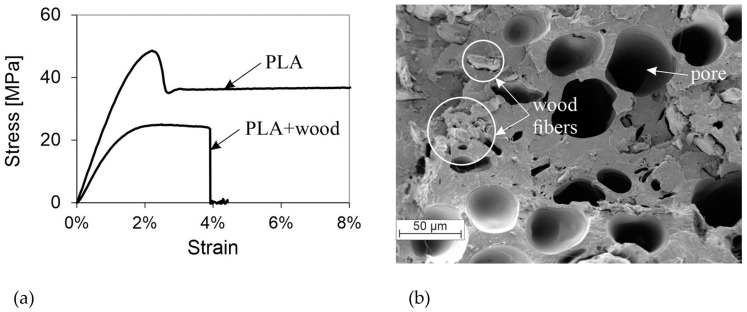
(**a**) Stress–strain curves of representative specimens from PLA and PLA + Wood; (**b**) SEM image showing the fractured surface of PLA filament with wood fibers.

**Figure 7 polymers-16-01576-f007:**
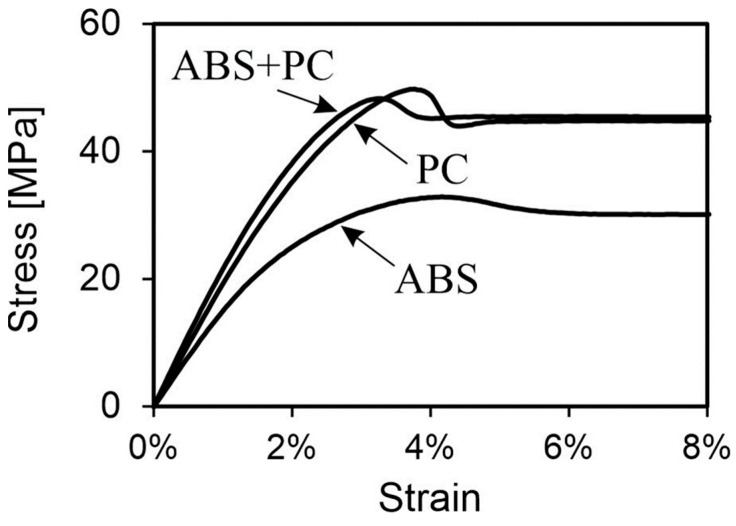
Stress–strain curve of representative specimens from ABS, PC and ABS + PC.

**Figure 8 polymers-16-01576-f008:**
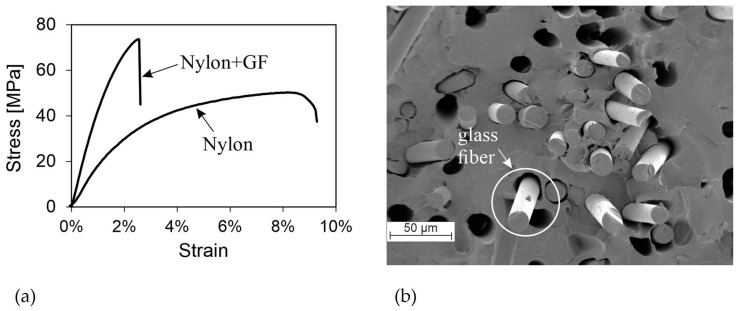
(**a**) Stress–strain curves of representative specimens from Nylon and Nylon + GF; (**b**) SEM image showing the fractured surface of Nylon filament with glass fibers.

**Figure 9 polymers-16-01576-f009:**
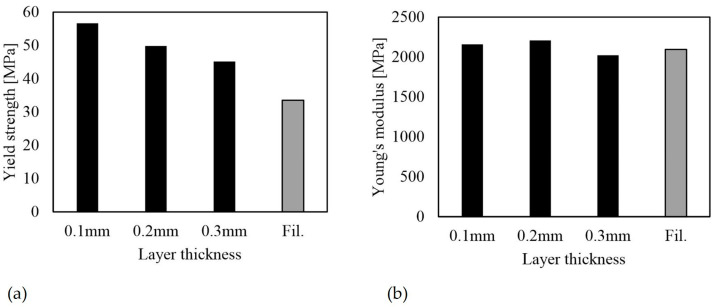
Mechanical properties of ABS + PC depending on the layer thickness of specimens A, B, and C as well as for the filament: (**a**) yield strength; (**b**) Young’s modulus.

**Figure 10 polymers-16-01576-f010:**
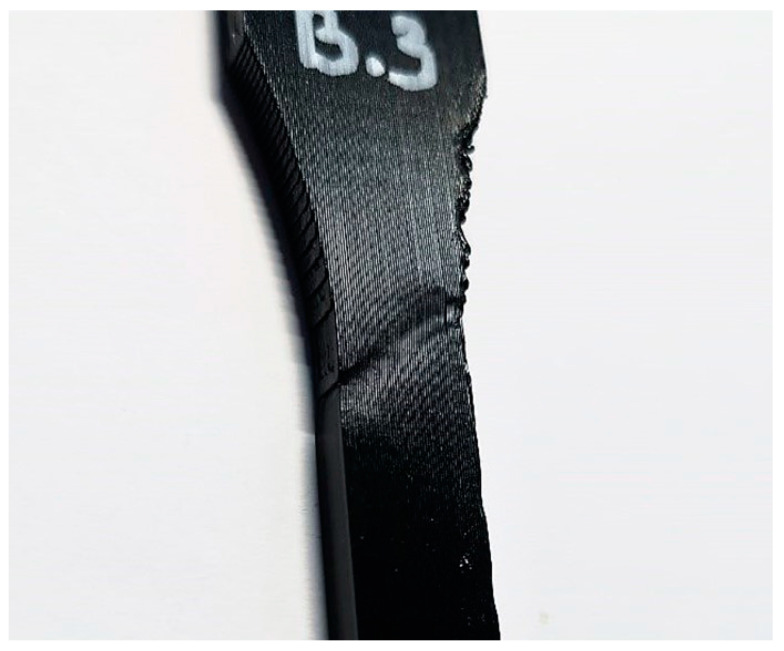
Necking in the taper area of specimen B.3.

**Figure 11 polymers-16-01576-f011:**
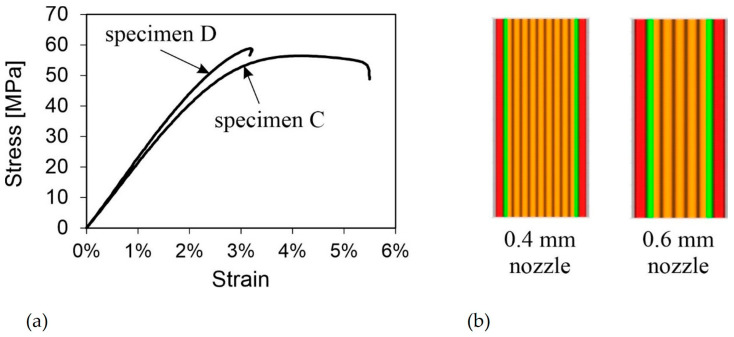
(**a**) Stress–strain curves of representative specimens C and D; (**b**) comparison of the inner layer structure of specimen with a 0.4 mm nozzle and a 0.6 mm nozzle.

**Figure 12 polymers-16-01576-f012:**
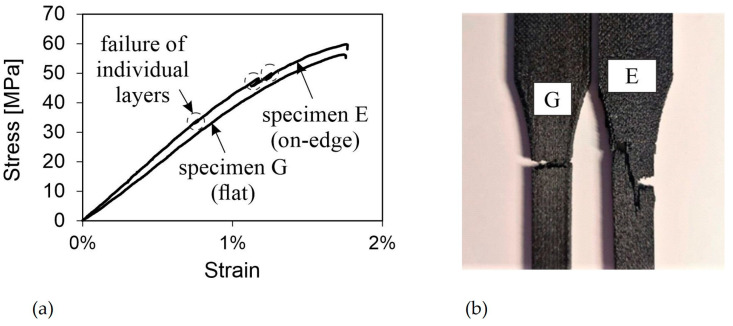
(**a**) Stress–strain curves of representative specimens E and G; (**b**) Fracture patterns of specimens E and G.

**Figure 13 polymers-16-01576-f013:**
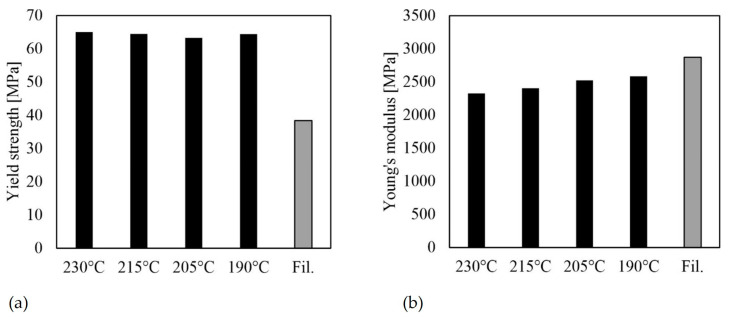
Mechanical properties for different nozzle temperatures and for filament for PLA (average values and standard deviations): (**a**) yield strength; (**b**) Young’s modulus.

**Figure 14 polymers-16-01576-f014:**
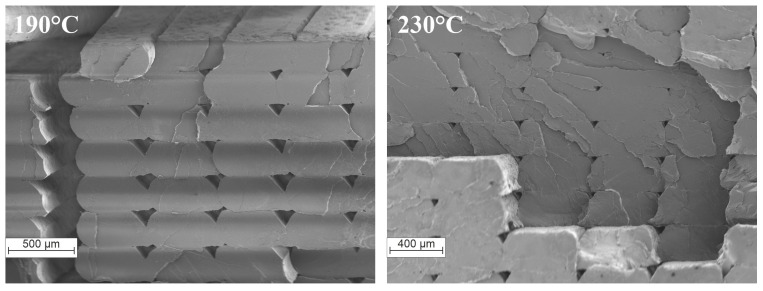
SEM images of PLA specimens printed at 190 °C (**left**) and 230 °C (**right**).

**Figure 15 polymers-16-01576-f015:**
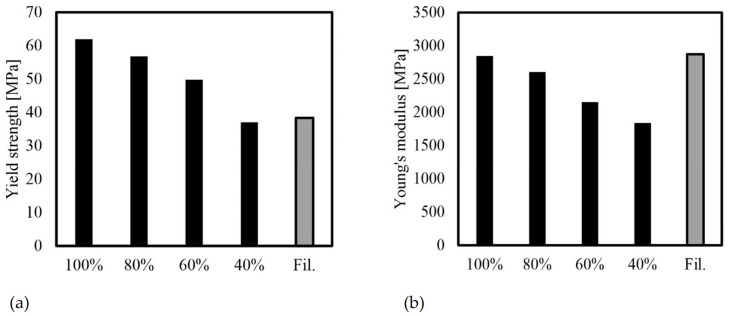
Mechanical properties for different infill densities and for filament (average values): (**a**) yield strength; (**b**) Young’s modulus.

**Figure 16 polymers-16-01576-f016:**
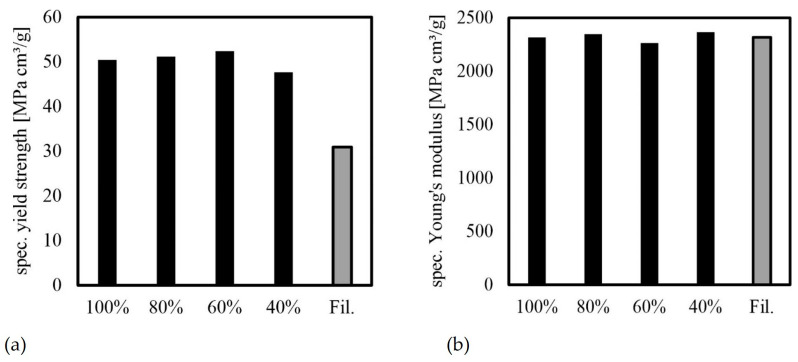
Specific mechanical properties for different infill densities and for filament (average values): (**a**) yield strength; (**b**) Young’s modulus.

**Figure 17 polymers-16-01576-f017:**
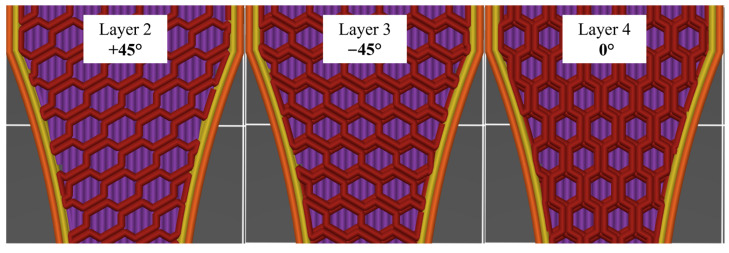
Structure of the honeycomb filling pattern.

**Figure 18 polymers-16-01576-f018:**
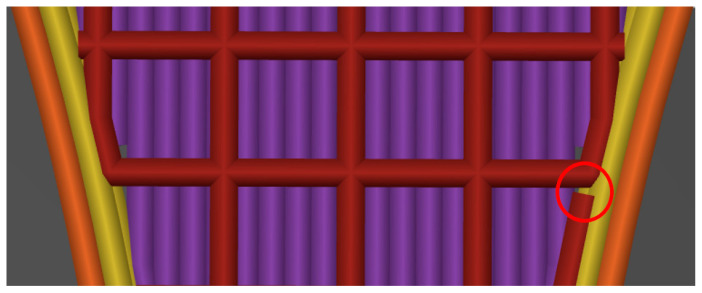
Unconnected grid structure.

**Figure 19 polymers-16-01576-f019:**
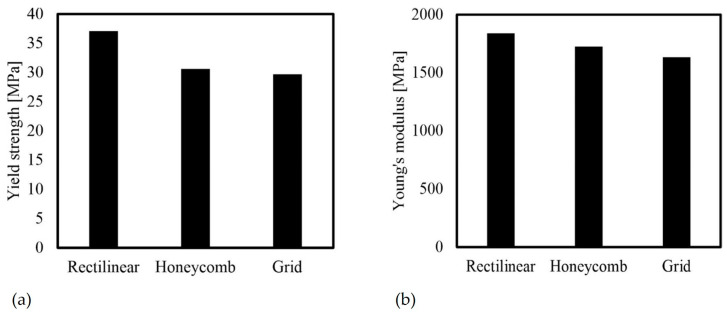
Mechanical properties for different infill patterns at 40% infill density (average values): (**a**) yield strength; (**b**) Young’s modulus.

**Figure 20 polymers-16-01576-f020:**
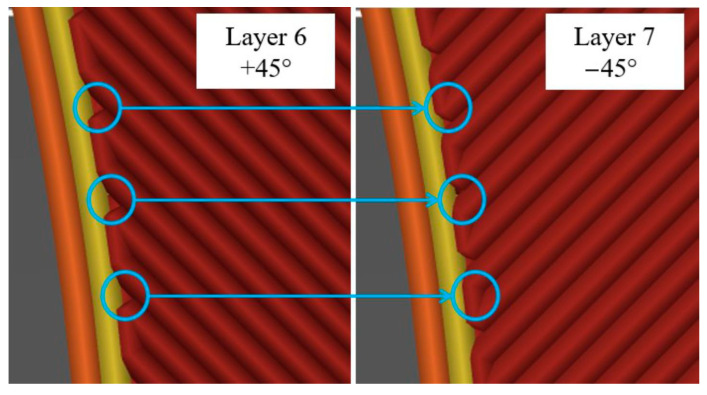
Gaps at ±45° raster angle for specimen S.

**Figure 21 polymers-16-01576-f021:**
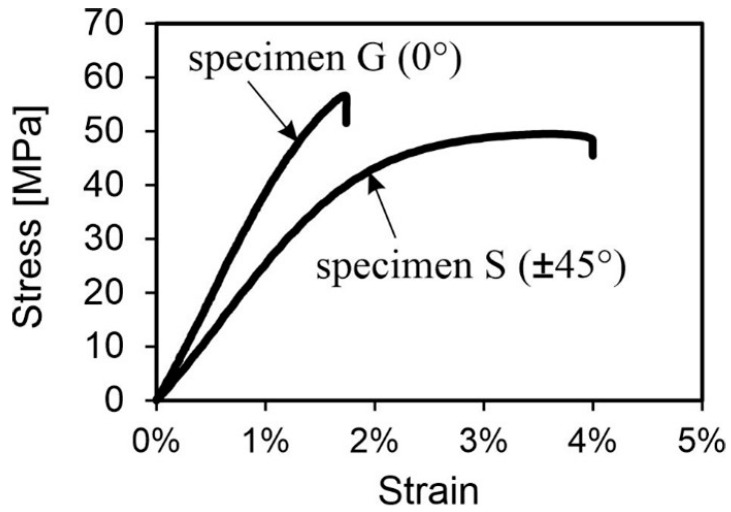
Stress–strain curves of representative specimens G and S.

**Figure 22 polymers-16-01576-f022:**
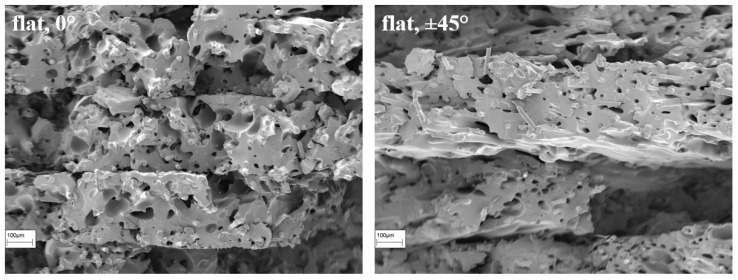
SEM images of Nylon + GF specimens for 0° (**left**) and for ±45° (**right**) raster angle.

**Figure 23 polymers-16-01576-f023:**
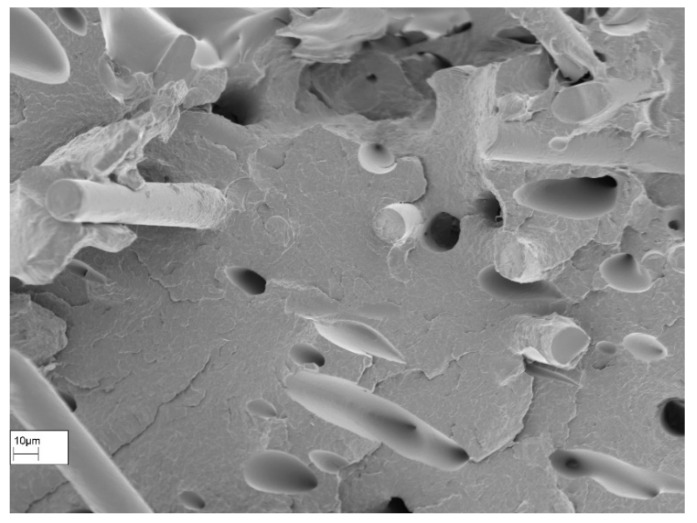
SEM image of Nylon + GF specimen for ±45° raster angle at higher resolution.

**Table 1 polymers-16-01576-t001:** Material overview (data according to manufacturers).

		PLA	PLA + Wood	ABS	PC	ABS + PC	Nylon	Nylon + GF
Properties	Unit							
Trade name	-	Prusament PLA	woodFill	Ultimaker ABS	PolyMax PC	ABSpro	Adura	FibreX PA6 + GF30
Manufacturer	-	Prusa Research	Colorfabb	Ultimaker	Polymaker	Formfutura	AddNorth	3DX Tech
City		Prague, Czech Republic	Belfeld, the Netherlands	Utrecht, the Netherlands	Changshu, China	Nijmegen, the Netherlands	Ölsremma, Sweden	Michigan (MI), USA
Price per 100 g	EUR	2.48	5.37	5.13	5.99	7.58	8.98	12.00
Diameter	mm	1.75	1.75	2.85	2.85	2.85	2.85	2.85
Material properties								
Density	g/cm^3^	1.24	1.15	1.10	1.18–1.20	1.18	1.10	1.35
Fiber material	-	-	Wood	-	-	-	-	Glass
Hygroscopy	-	Low	Low	Low	High	High	High	High
Fiber content	-	-	30%	-	-	-	-	30%
Mechanical properties								
Test standard	-	ISO 527	ISO 527	ISO 527	ASTM D638	ISO 527	ISO 527	ISO 527
Tensile strength	MPa	50	46	39	60	65.1	50	62.8
Tensile modulus	MPa	2.300	3.290	1.682	2.048	2.440	1.720	4.261
Elongation at break	%	2.70	4.80	4.80	12.24	6.60	46	6
Process requirements								
Nozzle requirements	-	no specifications	>0.4 mm	no specifications	no specifications	no specifications	no specifications	steal nozzle, >0.4 mm
Nozzle temperature	°C	200–220	195–220	250	250–270	240–260	245–270	220–280
Build plate temperature	°C	40–60	50–60	80	90–105	110	>50	80–110
Source		[24]	[25]	[26]	[27]	[28]	[29]	[30]

**Table 2 polymers-16-01576-t002:** Experimental plan for testing printed specimens.

ID	Material	Orientation	Layer Thickness (mm)	Infill Density	Infill Pattern	Raster Angle	Nozzle Diameter (mm)	Nozzle Temperature (°C)
A	ABS + PC	On-edge	0.1	100%	Rectilinear	0°	0.4	260
B	ABS + PC	On-edge	0.2	100%	Rectilinear	0°	0.4	260
C	ABS + PC	On-edge	0.3	100%	Rectilinear	0°	0.4	260
D	ABS + PC	On-edge	0.3	100%	Rectilinear	0°	0.6	260
E	Nylon + GF	On-edge	0.3	100%	Rectilinear	0°	0.6	260
F	PLA	Flat	0.3	100%	Rectilinear	0°	0.6	215
G	Nylon + GF	Flat	0.3	100%	Rectilinear	0°	0.6	260
I	PLA	Flat	0.3	100%	Rectilinear	0°	0.6	215
J	PLA	Flat	0.3	100%	Rectilinear	0°	0.6	230
L	PLA	Flat	0.3	100%	Rectilinear	0°	0.6	205
M	PLA	Flat	0.3	100%	Rectilinear	0°	0.6	190
N	PLA	Flat	0.3	40%	Rectilinear	0°	0.6	215
O	PLA	Flat	0.3	60%	Rectilinear	0°	0.6	215
P	PLA	Flat	0.3	80%	Rectilinear	0°	0.6	215
Q	PLA	Flat	0.3	40%	Honeycomb	-	0.6	215
R	PLA	Flat	0.3	40%	Grid	-	0.6	215
S	Nylon + GF	Flat	0.3	100%	Rectilinear	±45°	0.6	260

**Table 3 polymers-16-01576-t003:** Process parameters and their corresponding values.

Parameter	Value
Layer thickness (mm)	0.1, 0.2, 0.3
Nozzle diameter (mm)	0.4, 0.6
Orientation	Flat, On-edge
Nozzle temperature (°C)	190, 205, 215, 230, 260
Infill density	40%, 60%, 80%, 100%
Infill pattern	Rectilinear, Grid, Honeycomb
Raster angle	0°, ±45°

**Table 4 polymers-16-01576-t004:** Results of filament testing (average of three specimens with corresponding standard deviations).

Material	Yield Strength σ_y_ (MPa)	Young‘s Modulus E (MPa)
PLA	38.4 (4.2)	2.873 (332)
PLA + Wood	19.0 (1.1)	1.827 (65)
ABS	19.6 (4.0)	1.461 (240)
PC	37.3 (9.2)	1.757 (247)
ABS + PC	33.5 (4.5)	2.095 (123)
Nylon	26.8 (2.3)	1.794 (111)
Nylon + GF	46.4 (6.4)	4.664 (301)

**Table 5 polymers-16-01576-t005:** Results of printed specimen testing (with their corresponding standard deviations).

ID	Tensile Strength σ_u_ (MPa)	Yield Strength σ_y_ (MPa)	Elongation at Break (%)	Tensile Modulus E (MPa)
A	66.3 (0.3)	56.7 (3.8)	4.62 (0.48)	2.159 (93)
B	61.0 (0.4)	49.8 (1.6)	4.03 (0.19)	2.208 (43)
C	56.1 (0.5)	45.1 (3.3)	5.50 (-)	2.022 (74)
D	57.7 (1.0)	44.9 (0.4)	3.12 (0.11)	2.340 (32)
E	60.3 (1.3)	35.0 (2.5)	2.03 (0.29)	4.339 (196)
F	63.5 (0.5)	61.9 (1.4)	2.48 (0.08)	2.845 (91)
G	56.9 (0.7)	54.2 (1.1)	1.78 (0.06)	3.647 (109)
I	64.5 (2.8)	64.5 (2.8)	2.68 (0.22)	2.405 (237)
J	65.0 (2.5)	65.0 (2.5)	2.74 (0.25)	2.328 (258)
L	63.3 (0.5)	62.9 (0.5)	2.60 (0.05)	2.524 (155)
M	64.4 (1.1)	63.0 (0.7)	2.76 (0.23)	2.587 (87)
N	39.0 (0.1)	37.1 (0.7)	2.35 (0.03)	1.839 (21)
O	50.2 (0.5)	49.8 (0.3)	3.85 (0.45)	2.152 (131)
P	58.2 (0.3)	56.8 (0.7)	2.54 (0.20)	2.607 (47)
Q	32.8 (0.2)	30.6 (0.6)	2.46 (0.08)	1.725 (42)
R	29.7 (0.6)	29.7 (0.6)	1.79 (0.04)	1.634 (36)
S	50.8 (1.1)	42.6 (1.3)	3.83 (0.11)	2.419 (41)

**Table 6 polymers-16-01576-t006:** Different specimen densities at different infill densities.

ID	Infill Density	Average Weight (g)	Volume (cm^3^)	Density (g/cm^3)^
F	100%	12.53	10.204	1.23
P	80%	11.33	10.204	1.11
O	60%	9.7	10.204	0.95
N	40%	7.93	10.204	0.78
Filament	-	-	-	1.24

## Data Availability

The raw data supporting the conclusions of this article will be made available by the authors on request.

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
