# Peer review of "Mechanical Properties of Raw Filaments and Printed Specimens: Effects of Fiber Reinforcements and Process Parameters"

_polymers, 2024, doi:10.3390/polym16111576_

Round 1
Reviewer 1 Report
Comments and Suggestions for Authors
1. In the abstract, “The first test series revealed that the addition of wood fibers significantly worsened the mechanical behavior of PLA…”. You need to clarify the wood fibre content. For example, Is it true if 5% fibres are added?
2. You must mention those “seven different materials” in the abstract.
3. Emphasize the novelty of the work in the final paragraph of the introduction section.
4. The final paragraph of the introduction section should summarize the actions taken, the methods employed, and the key findings of the study.
5. What is the merit, and motivation for studying the specific composites?
6. There is nothing about the sources of studied materials. Did you make those filaments? Did you purchase them? If so, where, product code, country, city, etc?
7. Page 4, line 167: “PLA with wood fibers, both of which are widely used in FDM due to their simple printability”. Unlike what you stated, 3D printing of PLA reinforced with wood fibres is not a common material for FDM.
8. Page 4, line 167: “..their simple printability and relatively good mechanical properties”. It is against what you stated in the abstract that “…wood fibers significantly worsened the mechanical behavior of PLA”. Explain.
9. All mentioned polymers in this manuscript must be in full form name for the first use.
10. Why did you use two different printers?
11. There is nothing about the fibres in the polymers to reinforce the filaments. Fibre length, Fibre diameter, etc.
12. Page 5, line 189: What do you mean by “unaffected material properties of the materials”?
13. Page 5, line 190: “To achieve this, the materials are tested in their filament form”. Why did you test them? You already had the mechanical properties.
14. What test procedure/standard did you employ to test the filaments?
15. Why did you print some samples on their edge? What were you looking for?
16. How did you select the parameter levels? For example, two series had 0.1mm and 0.2mm layer heights, while the rest of them had a 0.3mm layer height.
17. You stated that the raster angle is one of your parameters, but only one series is printed in a different raster angle.
18. What is the point of Table 3? Delete it.
19. When you talk about recycled materials, you must provide details regarding those recycled materials. There is nothing about the recycled materials in this research.
20. It seems like the test matrix for the printed specimens is not complete and does not provide a comprehensive test plan to investigate the effect of different parameters on the mechanical properties.
21. Was there a difference between the manufacture datasheet for the filaments and what you tested here? Compare and explain the differences/similarities.
22. There is much research recently published regarding 3D printing and filament making of waste and recycled materials in detail such as the below ones. Review and summarize them in the introduction section and show the novelty of your work compared to those works.
Circular economy innovation: A deep investigation on 3D printing of industrial waste polypropylene and carbon fibre composites
Sustainable fabrication of 3D printing filament from recycled PET plastic
Recovery of Particle Reinforced Composite 3D Printing Filament from Recycled Industrial Polypropylene and Glass Fibre Waste
Reviewer 2 Report
Comments and Suggestions for Authors
Reviewer comments
Manuscript ID: polymers-3014102
Title: Mechanical properties of raw filaments and printed specimens: Effects of fiber reinforcements and process parameters
Journal: Polymers
Authors delve into the examination of mechanical properties of FDM (Fused Deposition Modeling) components, focusing on the various factors that influence these properties both in terms of material and process-related variables. Specifically, they explore composites comprising diverse polymer matrices reinforced with different fiber materials to identify those with optimal properties. Also, they analysis the mechanical behaviors of these materials and how process variables affect them.
While the paper is well-aligned with the thematic scope of the journal, there are areas that require improvement. Therefore, I suggest the following revisions to enhance the quality and depth of the manuscript:
1. The paper is well written, and the experiments were well conducted. However, what is the novelty? The novelty of this article should be emphasized and clearly addressed in the introduction, highlighting the developments, findings, or improvements compared to similar research previously published.
2. Pages 4-5: Table 1: Regarding the selection of composite materials with different types of fiber, it appears that the objective of the work is unclear. The use of wood and glass fibers as inclusions, as well as different matrix materials such as PLA, ABS, and Nylon are employed, despite their differing mechanical properties, requires justification. The authors should explain in the abstract and introduction the concept behind choosing such a diverse range of composites and provide insight into why this variety was deemed necessary?
3. Pages 6-7: Table 2: The article would benefit from clarification on why specific material and FDM process parameter values were defined within the selected range. Understanding the reasoning behind these choices would enhance the comprehensibility of the research methodology.
4. Another factor that should be discussed concerning SEM images is the interaction between fibers and matrix and the adhesion between layers for each material. The authors should provide conclusions regarding this aspect.
5. It is interesting to analyze how the raster angle, orientation, and the process parameters affect each other, it would be better to more explain this point.
6. In section 3: Could you provide more information to enhance our understanding of delamination and to justify its effects within your printed specimens? Additionally, I would appreciate further insights into the aspect of delamination.
7. Table 5: How the value of the standard deviation in the tensile modulus is more significant than the tensile modulus itself.
8. Page 16: Line 487: You mention that “failure strains of the specimen remained relatively unchanged with the different infill density”. However, regarding specimen O, the failure strain is more significant than the other specimens. The authors should explain and discuss this effect in more detail with respect to the printing parameters. How did you set this parameter?
9. The conclusion should be shortened. The conclusion should be more concise and highlight only the main results.
10. Typos and grammar problems need to be corrected properly, authors should carefully check through the manuscript before submitting a revision.
The reviewer recommends that the author do major revision to the manuscript.
Comments on the Quality of English LanguageTypos and grammar problems need to be corrected properly, authors should carefully check through the manuscript before submitting a revision.
Round 2
Reviewer 1 Report
Comments and Suggestions for Authors
The paper is accepted in its current form.
Reviewer 2 Report
Comments and Suggestions for Authors
The authors have provided correct and comprehensive responses to the questions raised by the reviewer. In light of the revisions carried out, I recommend that the manuscript be accepted for publication.
Comments on the Quality of English Languageok